# Structural Adaptive, Self-Separating Material for Removing Ibuprofen from Waters and Sewage

**DOI:** 10.3390/ma14247697

**Published:** 2021-12-13

**Authors:** Anna Maria Skwierawska, Dominika Nowacka, Paulina Nowicka, Sandra Rosa, Katarzyna Kozłowska-Tylingo

**Affiliations:** 1Department of Chemistry and Technology of Functional Materials, Gdansk University of Technology, Narutowicza 11/12, 80-233 Gdansk, Poland; dnowacka97@gmail.com (D.N.); paulina.nowicka1234@gmail.com (P.N.); sandrarosaaaa@gmail.com (S.R.); 2Department of Pharmaceutical Technology and Biochemistry, Gdansk University of Technology, Narutowicza 11/12, 80-233 Gdansk, Poland; katarzyna.kozlowska-tylingo@pg.edu.pl

**Keywords:** adsorption, pharmaceutical pollutants, β-Cyclodextrin, sewage

## Abstract

β-Cyclodextrin nanosponge (β−CD−M) was used for the adsorption of ibuprofen (IBU) from water and sewage. The obtained material was characterized by scanning electron microscopy (SEM), Fourier transform infrared spectroscopy (FTIR), Brunauer–Emmett–Teller (BET), Barrett–Joyner–Halenda (BJH), Harkins and Jura t-Plot, zeta potential, thermogravimetric analysis (TGA), differential scanning calorimetry (DSC) and elementary analysis (EA). Batch adsorption experiments were employed to investigate the effects of the adsorbent dose, initial IBU concentration, contact time, electrolyte ions and humic acids, and sewage over adsorption efficiency. The experimental isotherms were show off using Langmuir, Freundlich, Hill, Halsey and Sips isotherm models and thermodynamic analysis. The fits of the results were estimated according to the Sips isotherm, with a maximum adsorption capacity of 86.21 mg g^−1^. The experimental kinetics were studied by pseudo-first-order, pseudo-second-order, Elovich, modified Freundlich, Weber Morris, Bangham’s pore diffusion, and liquid film diffusion models. The performed experiments revealed that the adsorption process fits perfectly to the pseudo-second-order model. The Elovich and Freundlich models indicate chemisorption, and the kinetic adsorption model itself is complex. The data obtained throughout the study prove that this nanosponge (NS) is extremely stable, self-separating, and adjusting to the guest structure. It also represents a potential biodegradable adsorbent for the removal IBU from wastewaters.

## 1. Introduction

Currently, worldwide concern about the presence of ibuprofen (IBU) and other pharmaceuticals in the environment is growing [1,2,3,4,5]. Drugs are released to the environment from various sources as parent compounds and metabolites. They are continuously introduced into the sewage from households, hospitals, production sites and others [6,7,8,9,10,11]. Their presence in surface waters and in drinking water has a negative impact on the environment and human health [3,12,13,14,15,16]. This is due to the fact that in wastewater treatment plants, active pharmaceutical ingredients (API) undergo only partial degradation [12,17,18,19] (Appendix A provides data on the content of the title drug in different waters).

Fortunately, there are methods to remove drugs from waters, which include adsorption, electrochemical techniques, photocatalytic degradation, ultrasonic processes, oxidation and membrane technologies [17,19,20,21,22,23]. Among these, adsorption is the most common [24,25,26,27,28,29]. Adsorption processes ensure the removal of both organic and inorganic pollutants. The most popular adsorbents are: graphene and its derivatives, zeolites, polymers [21,24,25,30,31,32] and activated carbons (AC) [27,33,34,35]. The last one has a very developed specific surface area and may allow even 100% efficiency of removing drugs from wastewater [36]. Nevertheless, this material has disadvantages, such as production under high-temperature conditions and unprofitable regeneration in the case of API [36,37].

A novel class of adsorbents is cyclodextrin nanosponges (NS). They consist of cyclodextrin (CD) and a cross-linking agent. CD are cyclic oligosaccharides with a truncated cone shape [38,39]. The arrangement of the hydroxyl groups makes the exterior of the torus hydrophilic and the interior hydrophobic [40]. In aqueous solutions, cyclodextrins include organic compounds containing hydrophobic structural elements forming supramolecular complexes [41,42,43,44]. The resulting structures are based mainly on weak van der Waals interactions and hydrogen bonds. In terms of the stoichiometry, the most common types of complexes are considered in CDs: guest molecules with a 1:1, 1:2, 2:1 and 2:2 ratio [45]. Good solubility of CD in water excludes their direct use in water treatment [46]; however, an appropriate modification consisting of obtaining a cross−linked structure makes this possible [47,48]. Depending on the volume of the CD cavities (α−CD 17.4; β−CD 26.2 and γ−CD 25.6 nm^3^) [49], various organic compounds can be captured from the aqueous solutions and released in a subsequent step by using a suitable solvent [50,51,52]. The advantages of such sponges are high adsorption efficiency compared to a small surface area, no toxicity, the possibility of multiple regeneration and reuse, modeling of specific interactions, and low cost of production and use [47,51,53,54,55]. Their positive features make them a promising material in removing pollutants from water. There are reports in the literature on the adsorption of dyes [41,56,57,58,59], pesticides [41,60,61,62], fluorocarbons [63,64], organic and inorganic compounds [56,65,66] from waters. The synthesis of NS is cost-effective, easy to control and safe for the environment [41,67]. A favorable aspect of using CD-materials is their easy biodegradation by fungi [68]. For example, native β−CD and its derivatives are widely used in the production of more stable and more soluble forms of nonsteroidal anti-inflammatory drugs (NSAIDs) such as IBU, diclofenac and ketoprofen. [69,70,71,72,73,74]. Nevertheless, the literature on the use of NS to remove the abovementioned drugs from aqueous solutions is sporadic [75,76,77,78,79]. The presented work fills this gap for IBU. For the first time, we present a material in the form of native cross-linked β-Cyclodextrin (β−CD−M) for removal of IBU from aqueous solutions and sewage.

## 2. Materials and Methods

### 2.1. Reagents and Techniques

Hexamethylene diisocyanate (1,6-HMDI ≥ 99%), β-Cyclodextrin (β−CD ≥ 97%), ibuprofen (drug meet USP testing specifications), and pyridine (anhydrous ≥ 99.8) were purchased from Sigma Aldrich (Poznan, Poland). Acetonitrile (HPLC grade) was supplied by Chempur (Poznan, Poland). Methanol (CH_3_OH ≥ 99.8%), sodium and calcium chlorides, as well commercial humic acids containing 50% of humic acids and 50% fulvic acids (HA, 90% dry matter) were purchased from POCH (Gliwice, Poland) and Agraplant (Kalisz, Poland), respectively. Potassium bromide (KBr spectroscopy grade) was delivered by Fisher Scientific (Warszaw, Poland) and dried before use. All chemicals were used without further purification. Hach cuvette tests (Poznan, Poland) were used to determine BOD_5_, COD, total nitrogen and phosphorus. Water purified by a Hydrolab-system was used to prepare the stock solutions (HLP-SPRING, Straszyn, Poland, temp. 22 °C, κ = 2.70 µS). The treated sewage was collected from a home sewage treatment plant operating in a system consisting of two settling tanks, a reed plot (first stage of treatment) and a plot planted with energy willow (second stage of treatment).

The final form of the adsorbent was obtained after drying material in a moisture analyzer (MA50/1.R, Radwag, Radom, Poland) and grinding in a ball mill (Pulverisette 7 classic line, Fritsch, Poznan, Poland). Fourier transform infrared (FT-IR) spectra were performed on a Thermo Nicolet iS10 (Warsaw, Poland) using the KBr pellet method. The spectral resolution was 4 cm^−1^ and the scanning range was from 400 to 4000 cm^−1^. In order to determine the adsorption mechanism, the interactions between β−CD and IBU were analyzed by means of one- and two-dimensional proton magnetic resonance. ^1^H NMR spectra were recorded in D_2_O on a Bruker Advance III HD 400 MHz spectrometer at 25 °C. The specific surface area was measured by the Brunauer–Emmett–Teller (BET) method [80], and pore size distribution (PSD) was measured using the classical Barrett–Joyner–Halenda (BJH) model [81] and the Harkins and Jura t-Plot method. The surface charge properties of the adsorbents were investigated by Zeta potential measurements, which were conducted at a different equilibrium pH using a Nano-ZS Zetasizer (Malvern Instruments Inc., Poznan, Poland). Thermogravimetric analysis (TGA) and differential scanning calorimetry (DSC) were performed on β−CD−M using a Mettler-Toledo TGA/DSC thermogravimetric analyzer. Samples were heated from 0 to 1000 °C at the rate of 10 °C/min under nitrogen or the air atmosphere (flow rate: 10 mL/min). The surface morphology β−CD−M was studied using scanning electron microscopy (SEM) on a Quanta FEG 250 scanning electron microscope (Poznan, Poland) operating at 10 kV. The elemental FLASH 2000 analyzer (Poznan, Poland) operating in the dynamic separation technique was used to measure the content of carbon, hydrogen and nitrogen. The oxygen content analysis was performed by the device in the pyrolysis mode.

### 2.2. Synthesis of β−CD−M

β−CD−M was synthesized following the procedure previously reported [82] with major modifications. To a flask containing 150 mL of dry pyridine, heated to 80 °C, 10 g of CD and 5.93 g of 1,6-HMDI were added simultaneously over two hours with vigorous stirring, which was continued for an extra 8 h. After cooling to room temperature, 300 mL of acetone was poured into the reaction mixture to precipitate the product. The precipitate was filtered after 24 h and air dried. To remove unreacted cyclodextrin, the precipitate was quantitatively transferred to a beaker with 200 mL of water and was stirred for 0.5 h. Next, the precipitate was again filtered and transferred to a beaker with stirring 200 mL of 0.1 M hydrochloric acid. The gradual release of pyridine was accompanied by an increase in pH, which has been corrected by adding a few drops of concentrated hydrochloric acid. After stabilizing the pH at 2, the precipitate was filtered off, rinsed with distilled water until the filtrate had a neutral pH, then the precipitate was allowed to dry at room temperature. The product was placed in a moisture analyzer and dried at 120 °C until constant weight. The thus dried product was ball milled, sieved and stored in a desiccator. The final reaction yield was 11.15 g. Detailed information on material analysis is presented in the Appendix A.

### 2.3. Determination of Grafted β−CD Able to Form Complex

The content of β−CD in the synthesized material was determined by the classical method based on phenolphthalein complexation [83,84,85,86]. The amount of that β−CD in the material (involved in complexation) was calculated using the following formula:(1)$$\beta -\mathrm{CD}\;\mathrm{grafed}\;\left(\%\right)=\frac{W_{\beta -\mathrm{CD}}}{W_{\beta -\mathrm{CD}-M}}\times 100,$$
where: $W_{\beta -\mathrm{CD}}\;$is the weight of the bound β−CD (mg) and $W_{\beta -\mathrm{CD}-M}$ is the weight of material (mg). Detailed information is presented in Appendix A.

### 2.4. Water Regain Analysis

Water regain (WR) is one of the important properties of materials for water treatment. It can be used to investigate the hydrophilicity of the adsorbent [48,87,88]. A total of 100 mg of β−CD−M was dispersed in deionized water for 24 h to reach the swelling equilibrium state. Then the wet polymer was drained through filter paper. The collected material was additionally blotted using dry filter paper and then weighed. These experiments were carried out with five replicates. Next, the water regain was calculated by Equation (2).
(2)$$Water\;regain\;\left(\%\right)=\frac{W_{w}-W_{d}}{W_{d}}\times 100,$$
where $W_{d}$ (mg) and $W_{w}$ (mg) represent the mass of material in dry and wet form, respectively.

### 2.5. Adsorption Procedure

#### 2.5.1. Analytical Method of Pollutant Concentration Determination

The concentrations of IBU in samples were determined by high pressure liquid chromatography (HPLC Agilent Technologies 1200 Series, Warsaw, Poland) using an ultraviolet detector. The analytical IBU detection method was obtained from the reported literature [89,90,91]. The samples were injected at a volume of 1 μL, and the flow rate of the mobile phase (using acetonitrile: acetate buffer pH 3.2, ratio 60:40) through the C−18 column (Zorbax Eclipse PAH, 4.6 × 100 mm^2^, 3.5 µm, Agilent Technologies, Warsaw, Poland) was set at 1.0 mL min^−1^. The concentration of IBU was measured by the UV detector at the wavelength of 220.8 (ref. 360 ± 100) nm.

IBU removal efficiency (RME) was determined by Equation (3),
(3)$$RME\;\left(\%\right)=\frac{C_{0}-C_{t}}{C_{0}}\times 100,$$
where the *RME* (%) was the efficiency of API removal, and $C_{0}$ (mg L^−1^) and $C_{t}$ (mg L^−1^) were the initial concentration and concentration at time *t* of the drug in the sample. In each case, at least three series of samples were prepared. Based on the obtained results, the means and standard deviations were calculated.

#### 2.5.2. Adsorption Kinetics in Batch Experiments

A total of 100 mg of β−CD−M was introduced into plastic vials containing 50 mL of an IBU solution (initial concentration of 20 mg L^−1^), respectively. The vials were shaken on a digital vortex mixer at 700 rpm at 25 °C. The supernatant liquids were sampled at different times (0, 1, 3, 6, 10, 20, 30, 40, 60 min) and filtered (Whatman Grade 540 filter paper). The amount of API bound to the adsorbent was determined by Equation (4),
(4)$$q_{t}=\frac{\left(C_{0}-C_{t}\right)V}{m},$$
where *m* (mg) is the adsorbent dosage and *V* (mL) is the solution volume used in the study. Because each model of adsorption kinetics analyzes different interactions between the solid phase and the components of the solution, seven models were selected for consideration. The linear models presented in Table 1 were used in further calculations, and linear regression analysis was used to evaluate the fit of the theoretical model.

#### 2.5.3. Adsorption Isotherms

The adsorption isotherm study was conducted to evaluate the adsorption capacity of the adsorbent. A total of 1 mg of the β−CD−M was added to 5 mL of IBU solutions (with initial concentration ranging from 10 to 100 mg L^−1^) and shaken on a digital vortex mixer at 700 rpm at 25 °C for 1 h. Then the supernatant liquid was sampled and filtered. The adsorption equilibrium concentration of each sample was measured by the HPLC. To gain satisfactory data on IBU adsorption as well as to obtain an optimal fit of the theoretical model, these experimental data were analyzed using two and three parameter isothermal equations (i.e., Langmuir, Freundlich, Halsey, Hill and Sips), and linear regression analysis was used to evaluate the fit of the theoretical model (Table 2).

#### 2.5.4. Thermodynamic Analysis

The key thermodynamic parameters of adsorption were determined on the basis of the following equations.
(5)$$\ln\left(K_{e}^{0}\right)=\frac{-∆H^{0}}{RT}+\frac{∆S^{0}}{R},$$
(6)$$∆G^{0}=-RTln\left(K_{e}^{0}\right),$$
(7)$$K_{e}^{0}=\frac{KM_{a}C_{a}^{0}}{\gamma },$$
where $∆H^{0},\;∆S^{0},∆G^{0}$ are standard enthalpy, entropy and free energy and $K_{e}^{0}$ is a dimensionless thermodynamic equilibrium constant, which is calculated from the best isotherm fitted *K* (L g^−1^) estimated at temp. 20–60 °C, $M_{a}$ and $C_{a}^{0}$ are molecular weight and standard concentration of the adsorbate (1 M), and *γ* is the coefficient of activity, respectively [104].

#### 2.5.5. Effects of the Dosage of the Adsorbent, pH, Humic Acid and Ionic Strength

The effect of the adsorbent dosage was conducted by addition of the β−CD−M (1–100 mg) to the plastic vials (8 mL) containing an IBU solution (5 mL, 20 mg L^−1^) and shaken on a digital vortex mixer at 700 rpm at 25 °C for 1 h. Then the supernatant liquid was sampled and filtered. The adsorption equilibrium concentration of each sample was measured by HPLC. The influence of ionic strength was investigated analogously, using drug solutions enriched with appropriate salts, humic and fulvic acids, and the pH was adjusted to the appropriate value in the range from 2 to 11 with 0.1 N HCl and 0.1 N NaOH.

#### 2.5.6. Sewage Treatment

The wastewater (Table 3) was used to make solutions containing a drug. Relatively high concentrations of IBU, compared to those determined in the natural environment, were used deliberately as a simulation of a situation where the compound is discharged directly into the sewage system. The adsorption procedure was the same as for model water samples with one difference, and the filtered wastewater obtained during the second biological treatment was used to prepare the solutions instead of pure water. The 5 mL of IBU (0.021, 0.206 and 2.06 mg L^−1^) solutions were added to plastic vials containing 10 mg of β−CD−M, individually. The vials were sealed, and the mixtures were shaken on a digital vortex mixer at 700 rpm for 1 h at 25 °C. Then the supernatant liquid was sampled and filtered. The adsorption equilibrium concentration of each was measured by the HPLC.

#### 2.5.7. Material Regeneration

A total of 100 mg of β−CD−M was added to 50 mL of a solution containing IBU (20 mg L^−1^). The mixture was shaken in a digital vortex mixer at 700 rpm (25 °C, 1 h). After five minutes, the solution was decanted from the surface of the precipitate and, in its place, 20 mL of methanol was added, soaking for 10 min. The procedure was repeated using distilled water. The adsorption–desorption experiment was performed five times.

## 3. Results and Discussion

### 3.1. Adsorbents Properties

The method of β−CD crosslinking was selected based on a literature report [82]. In the characteristics of the new material, the authors indicated a quite large specific surface area of 78.06 m^2^ g^−1^ in the case of using a fourfold excess of HMDI and an extraordinary selectivity of the substitution of the primary hydroxyl groups. We were concerned about the limited solubility of β−CD in dry pyridine, used as a solvent so, the procedure has been modified by introducing both reagents gradually. This change resulted in a completely different material. The elemental analysis of the material showed a higher than the assumed degree of the CD cross-linking (Appendix A). On average, five/six hydroxyl groups of the β−CD molecule were substituted. It can be expected that the structural element (basic unit) of the material is composed of three β−CD molecules. Assuming the preferred mechanism of primary alcohol substitution, a total of 21 groups are available in a single β−CD, 16 are substituted, of which 14 are involved in urethane bonds and the other two in allophanate structures. The presence of both types of groups was confirmed by FTIR analysis, as was the disappearance of acute overtone 1078.86 cm^−1^, belonging to OH stretching primary alcohol (Figure 1c and Appendix A). Taking the described system as the basic structural element of the adsorbent, it should be expected that the average β−CD content in the material is about 54%. This could easily be confirmed by colorimetric measurements using phenolphthalein as a model (Appendix A). The obtained average result was 55.1% for 10 mg of β−CD−M. It should be kept in mind that this method has significant limitations resulting from the range of β−CD concentrations used [83,105]. Nevertheless, the convergence of the results is quite substantial, and it suggests that most macrocycles are available and will be able to take part in the removal of drugs from aqueous solutions. The surface analysis of the obtained material showed that it is a completely different material from the one described earlier by Ding [82]. The BET surface area of the material is 7.31 m^2^ g^−1^, which is quite typical [106] (Appendix A), whereas the average pore size is 16.8 nm and the total pore volume is 3.08 mL g^−1^ (Appendix A). The mean particle size of β−CD−M is between 270 and 600 μm, estimated by SEM. The particles have a distinct porous structure, resembling a pumice stone (Figure 1e,f). As estimated by TGA analysis, the material is thermally stable below 250 °C, followed by its decomposition. The ultimate weight loss is observed at 333.13 °C (61.82%) in an air and 346.67 °C (55.46%) in a nitrogen atmosphere, respectively (Appendix A). The material is intended for the treatment of wastewater, the temperature of which, even in industrial conditions, is not higher than 40 °C; therefore, it should be considered as completely durable. The tested Zeta potential (ξ) of β−CD−M in the pH range from 2 to 11 is 0 for pH = 2.53, and above this value it is constantly negative, assuming the value of −15 mV for a neutral pH. The system oscillates at the border of the agglomeration threshold up to a pH of 9, achieving then moderate stability with slight dispersion (ξ = −21 to −28 mV) (Figure 2). The high degree of cross-linking increases the hydrophobicity of the material and the tendency to accumulate, making it extremely easy to separate and regenerate. Due to the large particle size, the adsorbent settles completely within five minutes after stopping mixing. This property is a great advantage of β−CD−M when treating post-production wastewater before it is sent to a biological treatment plant, as it enables simple separation techniques such as decantation to be used. During mixing, the material slightly crumbles. The resulting particles form agglomerates very quickly, which prevents material loss during sedimentation or filtration. The material does not permanently retain water. The conducted tests showed an increase in the mass of the adsorbent samples up to 20%. After 24 h at room temperature, the samples returned to their original weight (Figure 3). The material used for wastewater treatment in dry and wet form shows the same sorption capacity in relation to the tested IBU. The adsorbent has no tendency to swell, which additionally confirms the high degree of cross−linking of the material [67].

### 3.2. Adsorption Experiments

#### 3.2.1. Effect of Adsorbent Mass on the Removal Efficiency

This is quite typical of a gradual increase in sorption due to the increased weight of the adsorbent. This is usually the result of incomplete filling of the adsorbent surface, as stabilization occurs only after the material is saturated. Hence, the IBU adsorption mechanism must be different. In the range of adsorbent concentrations (2 g L^−1^), the RME parameter decreases in relation to the initial value (0.2 g L^−1^) and then increases finally reaching 90% (Figure 4a). The adsorption capacity of the material reaches its highest value when the minimum amount of the adsorbent is used (Figure 4b). A quite characteristic feature of cyclodextrin NS is the ability to quantitatively capture organic substances present in very low concentrations, which cannot be said about other adsorbents [107]. The initial pH of the API solutions (20 mg L^−1^) was not corrected.

#### 3.2.2. Effect of IBU Initial Concentration on the Removal Efficiency

Figure 5a shows the effect of an initial API concentration in the range of 1–100 mg L^−1^ for drug removal. The initial concentration is the driving force needed to overcome the resistance to mass transfer between the aqueous and solid phases. At the first moment of contact, the solid phase has the greatest number of centers capable of interacting with the components of the aqueous phase, so IBU is removed quantitatively from the most diluted solution (1 mg L^−1^). At higher concentrations, an additional barrier to overcome is in the form of API particles protruding from toruses causing steric hindrance or simply accumulating on the outer surface of β−CD−M. Therefore, the participation of deeper structures in the process is limited and, consequently, a reduction in IBU removal is observed. However, when considering the degree of material filling depending on the value of the initial concentration of IBU, an upward trend can be noticed with the maximum saturation falling for the concentration of 80 mg L^−1^ (Figure 5b). The concentration of 20 mg L^−1^ was selected for further research.

#### 3.2.3. Effect of Contact Time in the Adsorption Experiments

Time is a determining parameter describing adsorption. The time taken to reach the maximum degree of removal depends on the reaction rate required to establish equilibrium. The optimal time for IBU seems to be 60 min (Figure 6a,b). The noted value of the RME parameter is 48.7% and the equilibrium capacity is 4.82 mg g^−1^, respectively. The greatest changes in the amount of IBU retained in the solid phase are visible in the first 10 min of the process, during which 4.26 mg is adsorbed per gram of adsorbent, and in the next 50 min—another 0.56 mg. In the case of less concentrated solutions, the equilibrium is established within the first two minutes. The high dynamics of the process will potentially enable the material to be used in environmental and industrial conditions, where time is of particular importance.

#### 3.2.4. Effect of the pH on the Removal Efficiency

IBU is moderately pH-sensitive (Figure 7). The electrokinetic potential of the various scattered phases depends largely on the concentration of hydrogen ions. In the case of the tested material, the electrokinetic potential is zero at pH = 2.53. The highest degree of IBU removal is placed exactly in the range of pH 2–3. This is the area close to the complete disappearance of anticoagulant electrostatic repulsion. However, in order to restore normal adsorption conditions, the pH of the IBU solutions used in further tests was not corrected and amounted to 4.54.

#### 3.2.5. Effect of Salts, Humic and Fulvic Acids on the Removal Efficiency

The presence of salt affects the effective removal of all tested drugs. In the case of IBU, both sodium and calcium chloride increase the effectiveness of drug removal by several percent. The addition of humic and fulvic acid also increases the removal of IBU. Acids have the greatest positive effect on adsorption when used in a concentration of 10 mg L^−1^. In this case, the value of the RME parameter increases by 66%. The obtained preliminary results indicate that the tested material should work properly in the case of real samples (Figure 8).

#### 3.2.6. β−CD−M Regeneration

The presented material contains elements of various flexibility, enabling a better adaptation to the structure of the adsorbed drugs. β−CD−M has a structure shaped during synthesis and contact with the pyridine template. During adsorption, the material is reorganized and arranged according to the structure of IBU. The drug molecules are very efficiently and quickly desorbed with methanol. The low boiling point of alcohol and the lack of azeotropes with water makes the regeneration stage relatively cheap and waste-free. The described adsorbent probably remembers the shape of the API and does not return to its initial state, which explains the initial increase in IBU adsorption and further stability in the degree of drug removal in subsequent trials using regenerated material (Figure 1d and Figure 9; Appendix A).

#### 3.2.7. Adsorption of Pharmaceuticals Present in Biologically Treated Sewage of Municipal Type

Wastewater (W) from a household sewage treatment plant after the second stage of treatment was used for the research. The mixtures were subjected to initial filtration, and the obtained solutions were used to determine the basic parameters, which are summarized in Table 3. For obvious reasons, the actual sewage differed significantly in the content of carbon, nitrogen and phosphorus from artificial ones, which could significantly affect the IBU adsorption process. Preliminary studies have shown that IBU can be removed from the actual wastewater with very high efficiency from 84% to 100% for concentrations ranging from 2.06 to 0.021 mg L^−1^, respectively (Figure 10). The highest concentration of IBU in raw sewage, which was recorded in Scotland, was 0.028 mg L^−1^ (Appendix A) [108].

#### 3.2.8. Adsorption Isotherms

A nearly straight line usually represents the equilibrium between the adsorbed amount and the residue in the solution. This Giles adsorption isotherm belongs to class C, describing a constant division of the adsorbed substance between the surface layer and the solution [109]. The obtained isotherm (Figure 11) is difficult to unambiguously assign to a given class.

Adsorption isotherms are in the form of mathematical equations, describing the relationship between the number of adsorbed molecules and those remaining in the solution. They are pressure, temperature and pH-dependent. Adsorption isotherms exhibit the path of a substance from an aqueous medium to a solid phase in a state of equilibrium. Five models of equilibrium isotherms were used in the work (Table 2 and Table 4).

The Langmuir model assumes that monolayer adsorption of molecules can take place only in specific, determined places at homogeneous adsorption energies on the surface without migration of the adsorbate in the surface plane. The empirical model describes quite imaginary conditions, i.e., homogeneous adsorption, and excludes interactions and steric hindrance between adsorbed molecules.

The essential characteristic of the Langmuir isotherm is complemented by the dimensionless separation coefficient *R_L_* parameter, the values of which indicate the type of the isotherm. Thus, an irreversible isotherm is described by a coefficient value of zero, favorable (0 < *R_L_* <1), linear (*R_L_* = 1), and unfavorable (*R_L_* > 1). The calculated values of the parameter clearly indicate the privileged course of the process (Table 4).

The Freundlich isotherm is used to describe multilayer adsorption and inhomogeneous surfaces with non−uniform heat distribution of adsorption. The Freundlich equitation and its linear form are presented in Table 2. The value of the parameter n determines the quality of the adsorption process; when it is between 1 and 10, the process is favorable, and when it is less from 1, it is poor. The correlation coefficients for IBU are worse compared to the Langmuir and Sips models. Nevertheless, the determined values of the parameter n indicate a favorable adsorption process in the studied concentration range of IBU (Table 4).

The Halsey isotherm can be used in determining the multilayer adsorption of heterosporous materials. High values of the linear regression coefficient indicate a fairly good fit to the discussed model, confirming the existence of different pores.

The Hill model makes it possible to distinguish the type of bonds of various molecules with homogeneous surfaces through the *n_H_* constant, which is a measure of the molecular interactions. The assumption of this model is that adsorption is a cooperative phenomenon with the adsorbate having the ability to bind at one site on the adsorbent, which may affect other binding sites on the same adsorbent. Moreover, in this model, it is assumed that if the coefficient $n_{H}=1$, then the binding is non-cooperative, and when $n_{H}>1$, the binding has a positive cooperation and a negative one when $n_{H}<1$. The last one describes the interaction of the β−CD−M with the IBU [110].

The Sips model works well for localized adsorption without the adsorbate–adsorbate interaction. As $C_{e}$ approaches a low value, the Sips isotherm effectively reduces to Freundlich, while at high $C_{e}$, it predicts the sorption properties of the Langmuir monolayer. The Sips isotherm equation is characterized by a dimensionless heterogeneity coefficient, *n*, which can also be used to describe the heterogeneity of a system if *n* is between 0 and 1. When *n* = 1, the Sips equation becomes Langmuir and represents the homogeneity of the adsorption process. In this case, n=1.33 may indicate that the sorption of one (first) IBU molecule increases the sorption of more of the others, which may appear to be apparently contradictory to the information obtained from the Hill model. However, when it is recognized that this is not the result of complex formation with a few ligands, but only adsorbent preorganization followed by proper adsorption, everything becomes coherent.

IBU adsorption fits well to the Sips model *(*$R^{2}=0.9934$), which was also used to determine the highest value of the maximum adsorption capacity ($q_{max}$), which was 86.21 mg g^−1^. By comparing the $q_{max}$ value with the values given in Table 5, it can be seen that this is a promising result considering the low cost of synthesizing the material based on general availability, production on a large-scale raw material, the simplicity of regeneration and no reduction in IBU removal efficiency.

The thermodynamic parameters, determined using the van’t Hoff equation, showed the spontaneity of the adsorption process of IBU. Adsorption is an exothermic process leading to an increase in disorder at the solid–liquid interface. Negative Gibbs free energy values indicate that sorption is sporadic and beneficial. In this case, the low enthalpy value confirms the formation of supramolecular complexes based mainly on van der Waals forces. (Table 6).

### 3.3. Kinetic Models

The adsorption rate is an important parameter used to describe the adsorption process. Many applications, such as wastewater treatment and drug disposal, require a fast adsorption rate and a relatively short contact time. To investigate the removal of IBU from water, seven models were used (Table 1). The constants of all kinetic models were calculated and summarized in Table 7. As can be seen in Table 7, the correlation coefficient ($R^{2}$) obtained from the pseudo-second-order was 0.9994 and was greater than for the pseudo-first-order model (0.965). Calculated values of $q_{e}$ ($q_{e}$, cal) on the basis of these two models unequivocally showed that only the pseudo-second-order model gives full agreement with the experimental values of $q_{e}$ ($q_{e}$, exp). Transport of the adsorbate from the solution to the adsorbent surface takes place in at least two stages. The phenomenon can therefore be controlled by film or external diffusion, pore diffusion, surface diffusion and pore surface adsorption, or a combination of more than one process. This theory was confirmed by a non-linear course of the intramolecular diffusion model, which shows that there are two separate stages of the drug sorption on β−CD−M, i.e., external diffusion and intergranular diffusion. The adsorption stage is generally very fast for porous adsorbents versus external or internal diffusion, and it is known that an adsorption equilibrium can be reached within minutes in the absence of internal diffusion [116,117,118,119]. However, the complete kinetics of the drugs adsorption process were described by a pseudo-second-order model, and it was not possible to clearly distinguish the rate-limiting stage of the process. That can be the boundary layer (film) or the diffusion inside the solid surface particles (pores) of substances dissolved in the solution. In the case of the obtained results, the long time to achieve the adsorption equilibrium (15approx. 1 h) indicates that internal diffusion may dominate the general kinetics of adsorption. To confirm unequivocal information about the rate-limiting stage, the assumptions of Weber and Morris were used on the basis of the internal diffusion model [118]. This model describes the equation shown in Table 1, where $k_{id}$ is the intra particle diffusion rate constant and *C* is the layer thickness. The greater the value of *C*, the greater the boundary layer influence will be on the adsorption process. If the factor limiting the process rate is diffusion inside the particles, then the dependence of the *q_t_* function of the square root on the time $t^{1/2}$ is a straight line through the origin of the coordinate system, i.e., C=0. The deviation from linearity clearly indicates that the rate-limiting factor may be a boundary layer (film) controlled by the diffusion process. As a result of the linear extrapolation of the first part of the straight line and its intersection with the ordinate, a value was obtained that corresponds to the diffusion boundary layer of adsorption or film thickness, while the second linear part concerns the diffusion inside the pores. The values of $k_{id-1},\;k_{id-2}$ as well as $C_{1}$ and$\;C_{2}$ were calculated by determining the slope of the lines defined by the experimental points in two-time intervals; the first in the range of 0 to 10 min and the second in the rest range. The values of these parameters are given in Table 7. The values of constants of $k_{id-1}$ are greater than of $k_{id-2}$, which may indicate that the adsorption process is initially controlled by the diffusive boundary layer (film). The *C* parameter value points to a greater boundary layer effect in the IBU adsorption process. Analyzing the data, it can be noticed that there are two separate stages of adsorption of IBU. The sharp linear course of the first part of the process it is associated with a diffusive boundary layer (film), so-called external diffusion, external surface adsorption or internal mass transfer effect. The second stage describes the gradual adsorption, surface diffusion and adsorption on the pore surface. The analysis of the intramolecular diffusion rate model also showed that the process takes place in two stages. During the first ten minutes, the adsorbate diffuses through the solution to the outer surface of the adsorbent. No sorption on the outer surface is observed. The second part that determines the rate (step 2. 10 to 60 min) is the gradual adsorption of drugs. There is still the question of whether the observed phenomena correspond to physical or chemical sorption. For this purpose, the Elovich and modified Freundlich models were applied. Both in the whole range were linear, and the slope describing the modified Freundlich kinetic model well below the value of one confirms the chemisorption of IBU at the concentration of 20 mg L^−1^. Tests with fewer concentrated solutions only allowed to determine the fit to the pseudo-second-order model, due to the very fast saturation of the adsorbent.

### 3.4. Mechanism of Adsorption

β−CD−M can be compared to a honeycomb, the cells of which are β−CD molecules and whose backbone is elastic urethane and allophanate chains. Once placed in the IBU solution, the material is preorganized to allow the host to better suit the requirements of the guest molecules. The β−CD content in the structure of NS is 55.1%, which means that theoretically the maximum adsorption capacity may be as high as 100.15 mg g^−1^ (Figure 12). The actual result is lower and reached 86.21 mg g^−1^, which may mean that not all tori are involved in supramolecular complexes. It also means that physical sorption outside the β−CD cavities can be ruled out. By examining the interaction of β−CD with IBU in a D_2_O solution, it can be concluded that the drug is an example of an excellent β−CD guest (Figure 13a). It contains a flat ring, substituted with linear groups in positions 1 and 4, which facilitates the insertion of the molecule into the CD cavity while involving the functional groups in the stabilization of the system (Figure 13b). All aromatic and even aliphatic protons take part in the complex formation (Figure 13b). The most stable complex is described by the 1:1 stoichiometry (Figure 13c). The strongest interactions occur between the H-5 protons, much weaker with H-3 and additionally H-6 with β−CD. In this case, the inclusion of the guest molecule is possible from both the smaller and the larger periphery. However, the ibuprofen molecule is too large to fit completely into the β−CD cavity; therefore, the protruding part of the linear fragments interacts with the H-6 β−CD protons outside the torus (Figure 12 and Figure 13d). The elements protruding from the cavity may impede other molecules trying to lodge in free cells as evidenced by the negative cooperation found with the Hill model. Nevertheless, it should be noted that the specific surface of the material is efficiently used, for example, compared to active carbon, whose maximum sorption capacity is only 19.7 mg g^−1^ greater, with the surface area almost 88 times greater (Table 4).

## 4. Conclusions

The article presents the synthesis of material obtained in the reaction of native β−CD, much cheaper than hydroxypropyl-β-Cyclodextrin, with hexamethylene diisocyanate. The material is highly cross-linked, as evidenced by the lack of swelling and the rapid desorption of water from the surface.

The adsorbent is easy to separate from the mixture as it is decanted in just a few minutes, which is extremely valuable when used as a movable bed. The sorption capacity remains unchanged after the next regeneration cycles.

Despite the small specific surface, the material is able to remove ibuprofen from wastewater even in concentrations much higher than usually found in the environment, which makes it possible to use it in the treatment of municipal and industrial wastewater, even very acidic.

The adsorption process requires about an hour of contact of the material with wastewater containing IBU and is described by a pseudo-second-order kinetic model and a Sips isotherm with a maximum adsorption capacity of 86.21 mg g^−1^.

## Figures and Tables

**Figure 1 materials-14-07697-f001:**
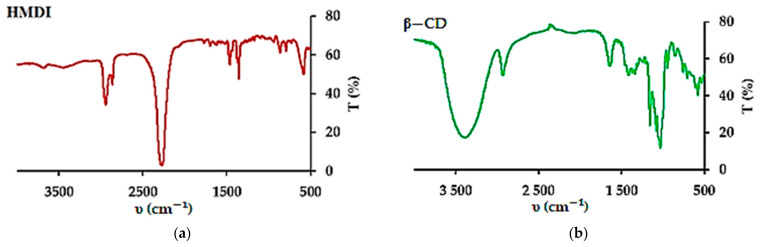
FTIR spectra (υ, KBr): (**a**) HMDI; (**b**) β−CD; (**c**) β−CD−M; (**d**) five times regenerated β−CD−M; (**e**,**f**) SEM images of β−CD−M.

**Figure 2 materials-14-07697-f002:**
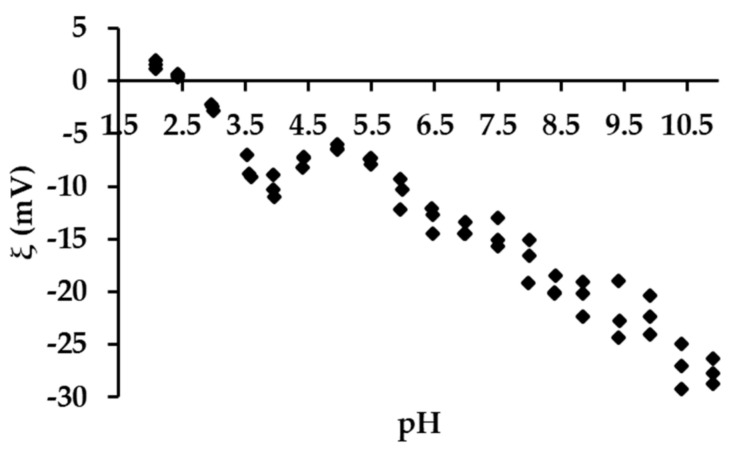
Zeta potential changes during titration of the range of pH 2–11.

**Figure 3 materials-14-07697-f003:**
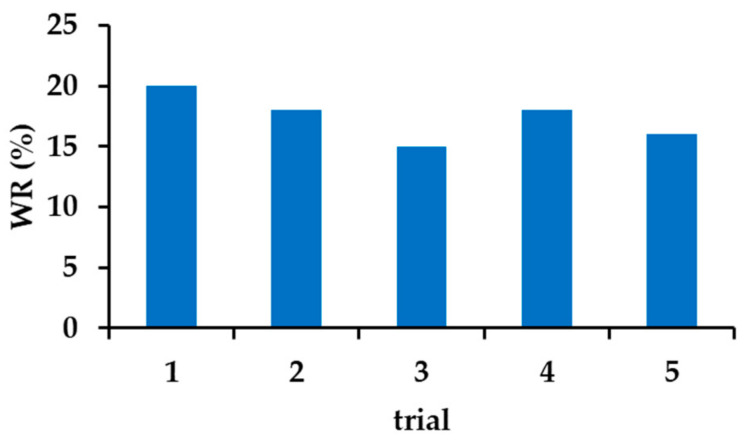
The amount of water retained in the β−CD−M.

**Figure 4 materials-14-07697-f004:**
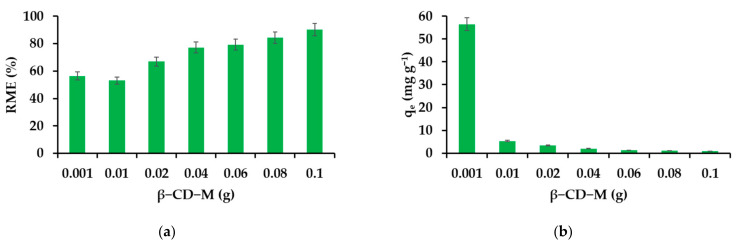
Effect of the adsorbent mass on the removal efficiency (**a**) and adsorption capacity (**b**).

**Figure 5 materials-14-07697-f005:**
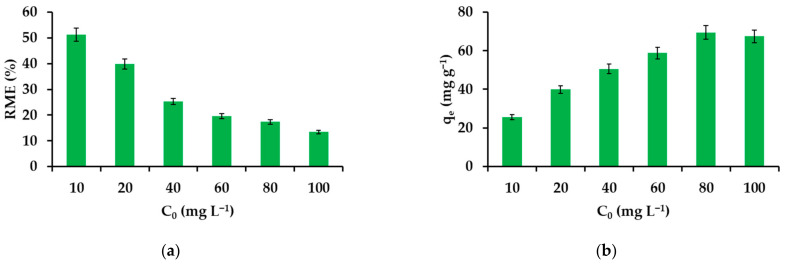
Effect of the initial IBU concentration on the removal efficiency (**a**) and adsorption capacity (**b**).

**Figure 6 materials-14-07697-f006:**
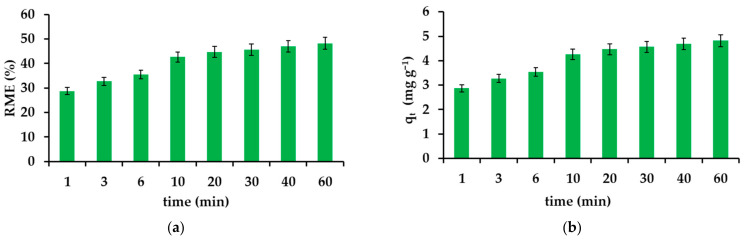
Effect of time on the removal efficiency (**a**) and adsorption capacity (**b**).

**Figure 7 materials-14-07697-f007:**
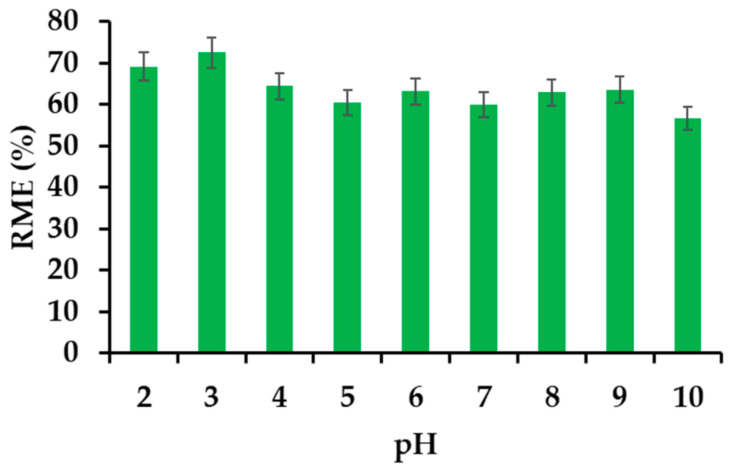
Effect of pH on the removal efficiency of IBU.

**Figure 8 materials-14-07697-f008:**
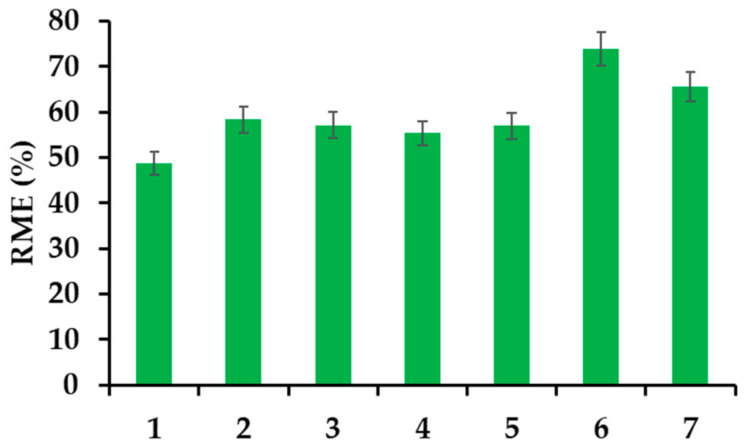
Effect of electrolytes and humic acids on the removal efficiency of IBU (20 mg L^−1^); (1) IBU in deionized water; (2) IBU in solution of NaCl (30 mg L^−1^) (3) IBU in solution of NaCl (300 mg L^−1^); (4) IBU in solution of CaCl_2_ (30 mg L^−1^) (5) IBU in solution of CaCl_2_ (300 mg L^−1^); (6) IBU in solution of humic acids (10 mg L^−1^) (7) IBU in solution of humic acids (30 mg L^−1^).

**Figure 9 materials-14-07697-f009:**
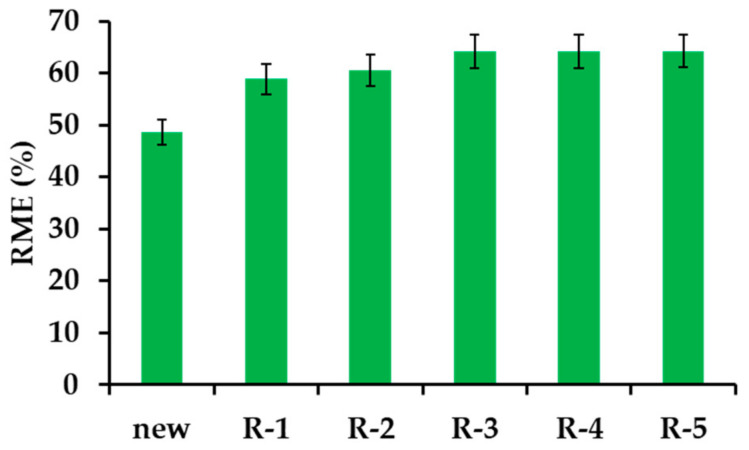
Comparison of the removal efficiency of IBU using a new and sequentially regenerated adsorbent.

**Figure 10 materials-14-07697-f010:**
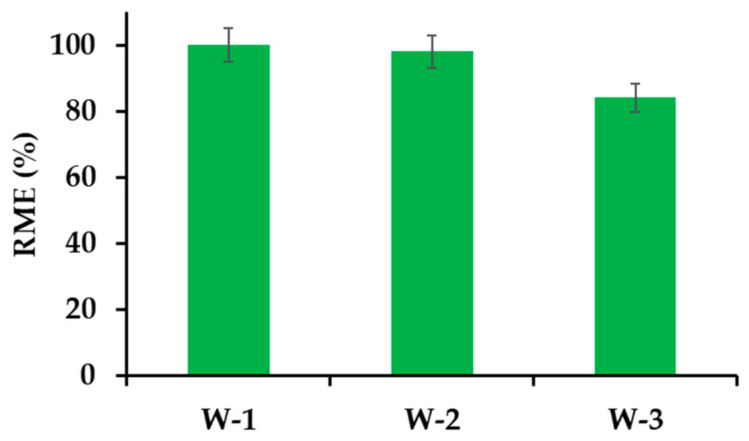
Effect of sewage containing different amounts of API (W-1, IBU 0.021 mg L^−1^), (W-2, IBU 0.206 mg L^−1^), and (W-3, IBU 2.06 mg L^−1^) on the removal efficiency.

**Figure 11 materials-14-07697-f011:**
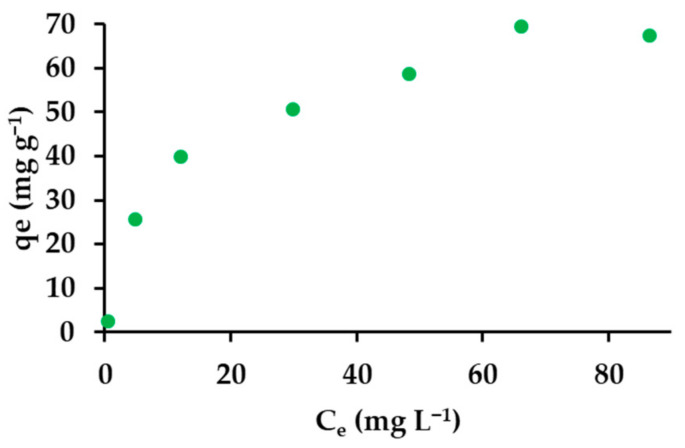
Equilibrium adsorption isotherm of IBU.

**Figure 12 materials-14-07697-f012:**
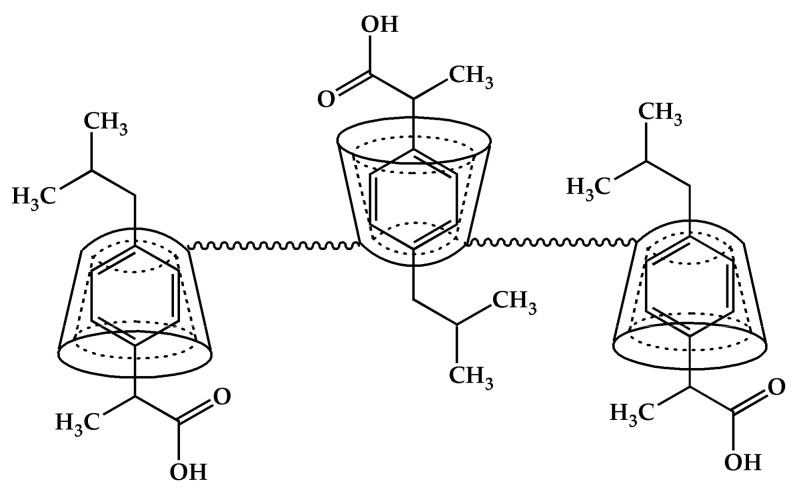
A probable fragment of a nanosponge structure containing absorbed IBU particles.

**Figure 13 materials-14-07697-f013:**
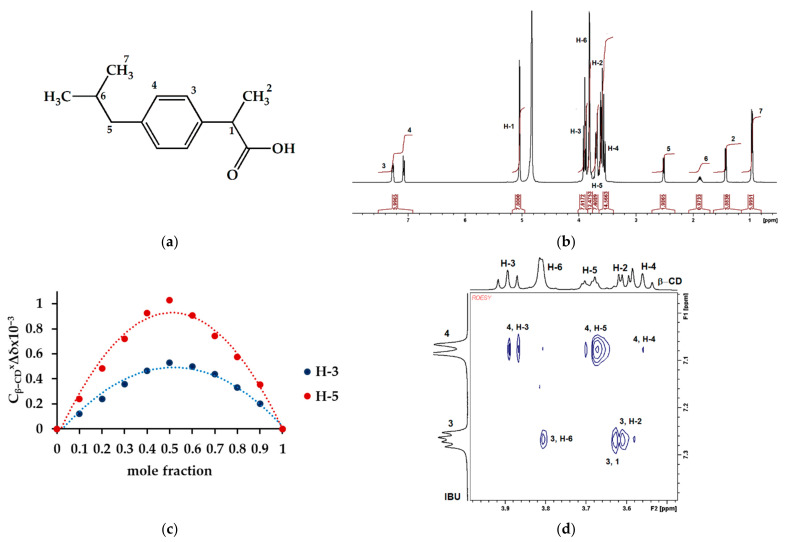
IBU complex with β−CD. (**a**) Chemical structure of IBU; (**b**) ^1^HNMR (δ, 400 MHz) spectrum of the complex registered in D_2_O; (**c**) complex stoichiometry determined on the basis of ^1^HNMR (δ, 400 MHz) spectra (continuous change method); (**d**) ROESY spectrum of the complex (1:1) in D_2_O.

**Table 1 materials-14-07697-t001:** Description of the models used in the calculation of kinetic constants and parameters.

Kinetic Model	Linear Form	Plot	Ref.
Pseudo-first-order	$\ln\left(q_{e}-q_{t}\right)=lnq_{e}-k_{1}t$	$\ln\left(q_{e}-q_{t}\right)\;\mathrm{vs}.\;t$	[92]
Pseudo-second-order	$\frac{t}{q_{t}}=\frac{1}{K_{2}}q_{e}^{2}+\frac{t}{q_{e}}$	$\frac{t}{q_{t}}\;\mathrm{vs}.\;t$	[20]
Weber and Morris	$q_{t}=K_{id}t^{1/2}+C$	$q_{t}\;\mathrm{vs}.\;t^{\frac{1}{2}}$	[93]
Liquid film diffusion	$\ln\left(1-\frac{q_{t}}{q_{e}}\right)=-K_{lf}t$	$\ln\left(1-\frac{q_{t}}{q_{e}}\right)\;\mathrm{vs}.\;t$	[94,95]
Bangham’s diffusion	$log\;\log\left\{\frac{c_{0}}{c_{0}-mq_{t}}\right\}=log\left\{\frac{mK_{B}}{2.303\;V}\right\}+\delta loq_{t}$	$log\;\log\left\{\frac{c_{0}}{c_{0}-mq_{t}}\right\}\;\mathrm{vs}.\;logt$	[96]
Elovich	$q_{t}=\frac{1}{\beta }\ln\left(\alpha \beta \right)+\frac{1}{\beta }\ln\left(t\right)$	$q_{t}\;\mathrm{vs}.\;\ln\left(t\right)$	[97]
Modified Freundlich	$lnq_{t}=\ln\left(k_{f}c_{0}\right)+\frac{1}{m}lnt$	$lnq_{t}\;\mathrm{vs}.\;lnt$	[98]

**Table 2 materials-14-07697-t002:** Models of adsorption isotherms and applied plots in the research.

Model *	Model in Linear Form *	Plot	Ref.
Langmuir	$q_{e}=\frac{q_{max}K_{L}C_{e}}{1+K_{L}C_{e}}$	$\frac{C_{e}}{q_{e}}=\frac{1}{q_{m}K_{L}}+\frac{C_{e}}{q_{m}}$	${\frac{C_{e}}{q}}_{e}\mathrm{vs}.\;C_{e}$	[99]
Freundlich	$q_{e}=K_{F}C_{e}^{\frac{1}{n}}$	$logq_{e}=lnK_{F}+\frac{1}{n}logC_{e}$	$logq_{e}\;\mathrm{vs}.\;logC_{e}$	[100]
Halsey	$q_{e}=exp\left(\frac{lnK_{H}-lnC_{e}}{n_{H}}\right)$	$lnq_{e}=\frac{1}{n_{H}}lnK_{H}-\frac{1}{n_{H}}ln\frac{1}{C_{e}}$	$lnq_{e}\;\mathrm{vs}.\;ln\frac{1}{C_{e}}$	[101]
Hill	$q_{e}=\frac{q_{sH}C_{e}^{n_{H}}}{K_{D}+C_{e}^{n_{H}}}$	$log\left(\frac{q_{e}}{q_{H}-q_{e}}\right)=n_{H}log\left(C_{e}\right)-log\left(K_{D}\right)$	$log\left(\frac{q_{e}}{q_{H}-q_{e}}\right)\;\mathrm{vs}.\;log\left(C_{e}\right)$	[102]
Sips	$q_{e}=\frac{q_{max}K_{s}C_{e}^{\frac{1}{n}}}{1+K_{s}C_{e}^{\frac{1}{n}}}$	$\frac{1}{q_{e}}=\frac{1}{q_{max}K_{S}}{\left(\frac{1}{C_{e}}\right)}^{\frac{1}{n}}+\frac{1}{q_{max}}$	$\frac{1}{q_{e}}\;\mathrm{vs}.\;{\left(\frac{1}{C_{e}}\right)}^{\frac{1}{n}}$	[103]

**Table 3 materials-14-07697-t003:** Characteristics of the sewage used in tests obtained from a home plant sewage treatment unit.

Parameter	Second Stage
BOD_5_ (mg L^−1^ O_2_)	30
COD (mg L^−1^ O_2_)	114
TOC (mg L^−1^ C)	2.2
Total suspended solids (mg L^−1^)	40
Total nitrogen (mg L^−1^)	10
total phosphorus (mg L^−1^)	0.5
Conductivity (μS cm^−1^)	1120
pH	6.2

**Table 4 materials-14-07697-t004:** Parameters of the equilibrium sorption models and of linear (*R*^2^) regression coefficients.

Isotherm Model	Parameters	IBU
Langmuir	*K_L_* (L mg^−1^)	0.082
	*q_max_* (mg g^−1^)	77.52
	*R* ^2^	0.990
$R_{L}=\frac{1}{1+K_{L}C_{0}}$	$R_{L}$	0.11–0.55
Freundlich	*K_F_* (L g^−1^)	15.823
	*n*	2.937
	*R* ^2^	0.9763
Halsey	*K_H_* (L g^−1^)	1.001
	*n_H_*	0.0003
	*R* ^2^	0.9763
Hill	*K_D_* (L g^−1^)	−0.994
	*n_H_*	0.0014
	*R* ^2^	0.9818
Sips	*K_S_* (L mg^−1^)	0.129
	*n*	1.33
	*q_max_* (mg g^−1^)	86.21
	*R* ^2^	0.9934

**Table 5 materials-14-07697-t005:** Maximum adsorption capacities for previously studied adsorbents for IBU removal from waters.

Adsorbent	Q_max_ (mg g^−1^)	Adsorption Conditions				Specific Surface Area (m^2^ g^−1^)	Reference
		pH	Temp. (°C)	Time (h)	C_0_ (mg L^−1^)		
Magnetic core-modified silver nanoparticles	0.29	7	25	0.75	0.2	116.48	[111]
Multiwall carbon nanotubes	1.15	4	25	2	200	151	[112]
Jordanian zeolite	1.23	2	25	1.33	510	41.20	[35,36]
Cocoa Shell biomass	21.43	2	30	10	30	no data	[113]
Acid-modified kola nut husk (KNHA)	28.22	not adjusted	30	3	20	712	[92]
Copper nanoparticles (Cu NPs)	33.9	4.5	25	1	10–40	no data	[114]
Chitosan-modified waste tire crumb rubber	70.00	6	25	24	40	no data	[27]
β−CD−M	86.21	4.54	25	1	1–100	7.31	this work
2−hydroxypropyl-β-Cyclodextrin polymers	87.50	5	25	2	50	no data	[75]
Activated Carbon	105.91	4	25	3	50	642	[9]
Zinc oxide sheet	220.00	no data	25	0.83	100	no data	[115]

**Table 6 materials-14-07697-t006:** Thermodynamic parameters for IBU adsorption on β−CD−M.

Temperature (K)	ΔH^0^ (kJ mol^−1^)	ΔS^0^ (J K^−1^ mol^−1^)	ΔG^0^ (kJ mol^−1^)
293			−19.32
313	−11.37	27.13	−19.83
333			−20.40
linear equation	value	SE	R^2^
intercept	3.2628	0.02007	0.9989
slop	1367.1	15.04890	

**Table 7 materials-14-07697-t007:** Kinetic model constants of single−component mixtures and their error analysis.

Isotherm Model	Intercept and Slop The Time Interval (min)	Parameters	IBU
First-order kinetic model	−0.0513	*q_e_* (mg g^−1^)	0.0513
	0.446	*k*_1_ (min^−1^)	1.56
	1–60	R^2^	0.965
Second-order kinetic model	0.2043	*q_e_* (mg g^−1^)	4.89
	0.3105	*k*_2_ (g min^−1^ mg^−1^)	77.162
	1–60	R^2^	0.9994
Bangham’s pore diffusion model	0.1647	δ	0.1647
	−3.2093	*K_B_* (mL L^−1^ g^−1^)	0.712
	1–10	R^2^	0.9936
	0.2588	δ	0.0702
	−3.466	*K_B_* (mL L^−1^ g^−1^)	0.905
	10–60	R^2^	0.9891
Weber and Morris	0.6165	C_1_	2.269
	2.2692	*K_id−_*_1_ (g mg^−1^ min^−1/2^)	0.616
	1–10	R^2^	0.9988
	0.109	C_2_	3.983
	3.9827	*K_id−_*_2_ (g mg^−1^ min^−1/2^)	0.109
	10–60	R^2^	0.9901
Liquid film diffusion	−0.1173 −0.7814 1–10	*K_lf_* (g mg^−1^ min^−1^)	0.117
	−0.0322 −0.5524	R^2^	0.9988
	10–60	*K_lf_* (g mg^−1^ min^−1^)	0.043
		R^2^	0.9896
Elovich	0.4936	β (g mg^−1^)	2.026
	0.9835	α (g mg^−1^ min^−1^)	176.080
	1–10	R^2^	0.9835
modified Freundlich	0.1364	1/m	0.136
	1.0731	*k_F_* (L g^−1^ min^−1^)	0.146
	1–10	R^2^	0.9767

## Data Availability

Not applicable.

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
