# Peer review of "Structural Adaptive, Self-Separating Material for Removing Ibuprofen from Waters and Sewage"

_materials, 2021, doi:10.3390/ma14247697_

Round 1

Reviewer 1 Report

Too many results and characterization techniques without plausible interpretation
The results in terms of maximum adsorption capacity do not justify the use of synthetic adsorbent with a method of obtaining which seems complicated despite the authors' insistence that it is an inexpensive method.
Authors should summarize the work and shorten the manuscript. A discussion of the results is more than necessary.
I think this manuscript needs rewriting with a good arrangement of results.

Author Response

Thank you very much for your reading and attention. The manuscript was redrafted taking into account the recommendations of all reviewers, which, unfortunately, were sometimes contradictory. Unfortunately, some comments seem unfounded to us.

Comment 1

‘Too many results and characterization techniques without plausible interpretation.’

Authors' reply

In our opinion, the presented interpretation of the results is credible. First, the material β-CD-M was characterized using standard techniques. It was possible to accurately characterize the structure of the polymer. Based on the FTIR analysis, the existence of urethane and allophanate groups was demonstrated. A high degree of cross-linking was established on the basis of elemental analysis. This technique also allowed to determine the basic structural units of the adsorbent, on the basis of a perfect match between the experimental and theoretical results, which is very difficult in the case of polymers (Anal. Calcd. (β-CD-M): C262H434N36O141: 7.95 (N), 49.60 (C), 6.89 (H), 35.56 (O). Anal. found. (β-CD-M): 7.93 (N), 49.26 (C), 6.84 (H), 35.05 (O) - S-2-1 in Supplementary Materials)

While investigating the kinetics of the IBU adsorption process, it was not only limited to determining the order of the reaction. An attempt was made to define the kinetic adsorption model of the process. The three models used showed that the kinetic of process is complex and takes place in two stages. Relatively good fit to the Elovic and Freundlich models showed that ibuprofen undergoes chemisorption. On the other hand, the values of the thermodynamic parameters ΔH0 = -11.37 kJ mol-1 do not confirm this. This discrepancy is easily explained by the formation of supramolecular complexes based mainly on Van der Waals forces, which are customarily attributed to physical adsorption and which are low in energy. The use of five adsorption models was aimed at a more profound analysis of the phenomenon, enabling the determination of the involvement of cyclodextrin toruses in the process and the space created by flexible polyurethane and allafoate chains. The obtained results excluded adsorption other than based on the formation of supramolecular complexes. The formation of IBU supramolecular complexes with β-CD has also been documented in the work.

Comment 2

‘The results in terms of maximum adsorption capacity do not justify the use of synthetic adsorbent with a method of obtaining which seems complicated despite the authors' insistence that it is an inexpensive method.’

Authors' reply

Currently, natural adsorbents are practically not used. Even those derived from renewable raw materials are processed before use. Cross-linked cyclodextrin, due to the presence of glucose, are biodegradable, which cannot be said about activated carbons, zeolites, titanium dioxide and other inorganic adsorbents. The interactions underlying the formation of supra-molecular complexes are weak, hence desorption is a fast and quantitative process. The material can be used many times. It separates easily and is sufficiently mechanically stable, which fully justifies its use. The value of the maximum adsorption capacity may not be the highest, but taking into account the concentration of IBU determined in ng level in untreated and treated sewage, rivers and other streams, it should be considered sufficient. Considering that the preliminary studies with real wastewater allowed the complete adsorption of IBU with an IBU concentration of 0.021 mg L-1

The synthesis is simple because it is one-step, does not require high temperature, use of inert gas or special equipment. In practice, the response time can be halved, as demonstrated by subsequent research. Indeed, the only troublesome part of the synthesis is the removal of unreacted β-CD and solvent.

Comment 3

Authors should summarize the work and shorten the manuscript.

Authors' reply

Whenever possible, the manuscript was shortened and the summary was further edited

Comment 4

A discussion of the results is more than necessary.

Authors' reply

Thank you for this remark. In the chapter entitled 'results and discussion' we tried to discuss each obtained result. Perhaps the discussion is not emphasized enough, which we tried to improve.

Comment 5

I think this manuscript needs rewriting with a good arrangement of results.

Authors' reply

The manuscript has been edited. We hope it made it more readable.

Reviewer 2 Report

The present article deals with the removal of ibuprofen from water. I will outline here some concerns that have to be discussed by authors.

I. The introduction is not gripping and can be improved. Additional information regarding the properties of CDs and their ability to remove pollulants can be included in this section.

II. Revise equation 1 and subscripts.

III Why molecules like DIC and KETO were included in lines 173-174?

IV. Methods employed to characterize the adsorbent are not included in section 2.

V. Personally, I don't like the organization of Figure 3. Too many graphs and it is difficult to follow the article.

VI. Did the authors analyze if results in Figure 3 are significant? Did the use any stastitical analysis to corroborate?

VII. Again, revise the appearence of Figure 4. Every single graph is different (size, legend, etc.)

VIII. If could be easier to follow table 4 if the order of articles is based on the qmax result. In these articles the adsorption of IBU is performed with different adsorbents?

IX. Can authors provide the slope and intercept of the Van't Hoff equation? Can authors provide the temperature used in Table 6?

X. Once more, the appearance of Figure 5 is not suitable at all. 

XI. Only 1 concentration was analyzed in Figure 5? Why?

XII. Reconsider the conclusion section is too long.

Author Response

Thank you very much for all your valuable comments, which undoubtedly enabled us to prepare a better version of the manuscript.

Comment I

The introduction is not gripping and can be improved. Additional information regarding the properties of CDs and their ability to remove pollutants can be included in this section.’

Authors' reply

Where possible, the introduction has been redrafted and supplemented with basic information on cyclodextrins. Unfortunately, another reviewer believes that the paper is too long and should be shortened, which is in contradiction to providing additional information. We hope that the introduction, despite everything in its current form, is more interesting.

Comment II

‘Revise equation 1 and subscripts.’

Authors' reply

We sincerely apologize for the editing error. The symbols in equation 1 are written correctly unlike those in the text which mistakenly do not contain the subscripts markup.

Line 129 was ‘where: Wβ-CD is the weight of the bound β-CD (mg) and WM is the weight of material (mg)’ and supposed to be ‘where: Wβ-CD is the weight of the bound β-CD (mg) and W β-CD-M is the weight of material (mg).’

Comment III

‘Why molecules like DIC and KETO were included in lines 173-174?’

Authors' reply

We sincerely apologize for the editing error. Of course, these letter abbreviations should not be in the text.

Comment IV

‘Methods employed to characterize the adsorbent are not included in section 2.’

Authors' reply

The second chapter is entirely devoted to the materials and methods. Indeed, the introduction of sub-chapter ‘2.1 Materials’ was incorrect, which has been corrected. Sections 2.3 to 2.5 describe all applied methods.

Comment V

‘Personally, I don't like the organization of Figure 3. Too many graphs and it is difficult to follow the article.’

Authors' reply

As requested by the reviewer, the grouped drawings were divided and transferred to the appropriate subchapters.

Comment VI

‘Did the authors analyze if results in Figure 3 are significant? Did the use any stastitical analysis to corroborate?’

Authors’ reply

All measurements were repeated at least three times. The reliability of the determinations was assessed on the basis of mean values and standard deviations.

Comment VII

‘Again, revise the appearence of Figure 4. Every single graph is different (size, legend, etc.)’

Authors’ reply

Figure 4 was removed from the manuscript except for point (a) because the data resulting from fitting the experimental points to the appropriate models are collected in Table 5 and Table 6.

Comment VIII

If could be easier to follow table 4 if the order of articles is based on the qmax result. In these articles the adsorption of IBU is performed with different adsorbents?

Authors’ reply

As suggested by the Reviewer, the data in Table 4 were ranked in order of increasing qmax value. Only one adsorbent in the form of cross-linked native β-cyclodextrin was investigated in the current study. The obtained test results were compared with other previously published ibuprofen adsorbents presented in Table 4.

IBU is quite a threat due to its endocrine properties and the fact that it is constantly introduced into the environment.It has even been found in drinking water.For this reason, various adsorbents are used to remove drug from aqueous solutions, as shown in Table 4.

Comment IX.

‘Can authors provide the slope and intercept of the Van't Hoff equation? Can authors provide the temperature used in Table 6’

As requested by the Reviewer, Table 6 supplemented with temperatures and data determined on the basis of the Van't Hoff equation.

Comment X.

‘Once more, the appearance of Figure 5 is not suitable at all.’

 Authors’ reply

Figure 5 was removed from the manuscript, at the same time table 7 was supplemented with data on the determined linear equations (intersection and slop).

 Comment XI.

‘Only 1 concentration was analyzed in Figure 5? Why?’

Authors’ reply

Lower concentrations of IBU (5 and 10 mg L-1) were also used in the studies, but the too short time (1-2 minutes) to reach the limit value of qe allowed only to establish that these are pseudo-second-order reactions. Finally, the concentration of 20 mg L-1 was selected, allowing for a more in-depth analysis of the studied phenomenon.

Comment XII.

‘Reconsider the conclusion section is too long.’

Authors’ reply

The conclusions have been redrafted.

Reviewer 3 Report

Title: Structural adaptive, self-separating material for removing ibuprofen from waters and sewage

Authors: Anna Skwierawska, Dominika Nowacka, Paulina Nowicka, Sandra Rosa and  Katarzyna KozÅ‚owska-Tylingo

General Comments:

  • In this study, the authors proposed for the first time a material in the form of native cross-linked β-cyclodextrin (β-CD-M) for removal of ibuprofen (IBU) from aqueous solutions and sewage.
  • The Introductory section is well organized. The novelty of this research is highlighted.
  • The authors present in the Materials and methods section the reagents and the equipment used, the synthesis of β-CD-M material, Determination of grafted β-CD able to form complex, Water regain analysis, and Adsorption procedure.
  • The synthesis method is described in detail in order to allow another researcher to reproduce the results.
  • In the Results and Discussion part, the authors present and interpret the results of the experiments performed. Also, the Mechanism of adsorption is described.
  • The paper ends with the main conclusions of their research study.

I suggest publishing the paper after minor changes:

  1. Section 2.5.3 – Please replace the reference ‘’Freundlich, 1906’’ with [87] in Table 2.
  2. Line 172-175 – Please verify the paragraph: ‘’ 1 mg of the β-CD-M was added to 5 mL of DIC, IBU and KETO solutions (with initial concentration ranging from 10 to 100 mg L-1) and shaken on digital vortex 174 mixer at 700 rpm at 25 °C for 1 h.’’
  3. Figure 3i -- I suggest to explain each abbreviation: Na-30, Na-300, Ca-30, Ca-300, Kw-10, Kw-30.
  4. Please check the numbering of the Section 3.2 taking into consideration that Section 3.2.8 is missing.
  5. In my opinion, Figure 4d ‘’ The van’t Hoff plots obtained for IBU sorption on β-CD-M contact time 60 min., temp. 20 °C, 40 °C and 60 °C’’ can be presented after the sentence: ’’ The thermodynamic parameters, determined using the van't Hoff equation, showed the spontaneity of the adsorption process of IBU (Fig.4g)’’ (Line 440).
  6. Indeed, the highest correlation coefficients were obtained in the case of Langmuir (R2= 0.990) and Sips isotherm (R2= 0.9934) compared with Freundlich (R2= 0.9763), Halsey (R2= 0.9763), and Hill (R2= 0.9818). Please verify the sentences below. Probably it is a type error:

Line 22-23 – In the Abstract section it is mentioned that: ‘’ The fits of the results are estimated according to the Sips isotherm, with a maximum adsorption capacity of 86.21 mg/g.

On the other hand, on the Line 327  it is noted that: ‘’ In the study case, the best representation seems to be the class L (Langmuir type).

Line 394- 396 -- IBU fits perfectly to the Langmuir and Sips models, which were also used to determine the highest value of the maximum of adsorption capacity (qmax), which was 77.52 mg g-1 and 86.21 mg g-1, respectively.

Line 575-577: ‘’The adsorption process requires about an hour of contact of the material with wastewater  containing IBU and is described by a pseudo-second kinetic model and a Sips isotherm  with a maximum adsorption capacity of 86.21 mg g-1.

Author Response

Thank you very much for your valuable comments and remarks that will make our manuscript better.

Comment 1.

‘Section 2.5.3 – Please replace the reference ‘’Freundlich, 1906’’ with [87] in Table 2.’

Authors’ reply

We sincerely apologize for our oversight, probably resulting from the use of initially different versions of the biography record. The citation has been changed to the correct one.

Comment 2.

‘Line 172-175 – Please verify the paragraph: ‘’ 1 mg of the β-CD-M was added to 5 mL of DIC, IBU and KETO solutions (with initial concentration ranging from 10 to 100 mg L-1) and shaken on digital vortex 174 mixer at 700 rpm at 25 °C for 1 h.’’

Authors’ reply

We sincerely apologize for the editing error. Of course, these letter abbreviations should not be in the text, the described adsorption method applies only to IBU.

Comment 3

‘Figure 3i -- I suggest to explain each abbreviation: Na-30, Na-300, Ca-30, Ca-300, Kw-10, Kw-30.’

Authors’ reply

The drawing has been corrected. The markings were changed to numbers, the meaning of which is explained in the figure caption.

Comment 4

‘Please check the numbering of the Section 3.2 taking into consideration that Section 3.2.8 is missing.’

Authors’ reply

Section numbering has been corrected.

Comment 5

In my opinion, Figure 4d ‘’ The van’t Hoff plots obtained for IBU sorption on β-CD-M contact time 60 min., temp. 20 °C, 40 °C and 60 °C’’ can be presented after the sentence: ’’ The thermodynamic parameters, determined using the van't Hoff equation, showed the spontaneity of the adsorption process of IBU (Fig.4g)’’ (Line 440).’

Authors’ reply

W fully agree with the Reviewer's opinion. However, taking into account the suggestions of all reviewers, this drawing has been removed from the manuscript and replaced with the description of the determined line equation, which is included in Table 6.

Comment 6

‘Indeed, the highest correlation coefficients were obtained in the case of Langmuir (R2= 0.990) and Sips isotherm (R2= 0.9934) compared with Freundlich (R2= 0.9763), Halsey (R2= 0.9763), and Hill (R2= 0.9818). Please verify the sentences below. Probably it is a type error:

Line 22-23 – In the Abstract section it is mentioned that: ‘’ The fits of the results are estimated according to the Sips isotherm, with a maximum adsorption capacity of 86.21 mg/g.

On the other hand, on the Line 327  it is noted that: ‘’ In the study case, the best representation seems to be the class L (Langmuir type).

Line 394- 396 -- IBU fits perfectly to the Langmuir and Sips models, which were also used to determine the highest value of the maximum of adsorption capacity (qmax), which was 77.52 mg g-1 and 86.21 mg g-1, respectively.

Line 575-577: ‘’The adsorption process requires about an hour of contact of the material with wastewater  containing IBU and is described by a pseudo-second kinetic model and a Sips isotherm  with a maximum adsorption capacity of 86.21 mg g-1.’

Authors’ reply

Indeed, the presented interpretation of the adsorption results is questionable. Referring to the comments of the reviewer, the discussion on the obtained adsorption results and the conclusions resulting from them were edited.

Round 2

Reviewer 1 Report

Authors have made the necessary corrections. I think it is now appropriate to proceed with the publication of the article.

Author Response

Dear Reviewer,

Please accept our thanks for valuable comments and time. The manuscript was linguistically checked again.

Yours faithfully

Anna Skwierawska 

Reviewer 2 Report

I. According to figure 11, the intercept of Van't Hoff is close to 7, however in the text is 3.26. Can authors clarify this point?

II. Figures 11 & 12 cannot be published in this form. The size of each graph is different. The graphs have to be homogeneized and ordered. Please include major ticks.

Author Response

Dear Reviewer,

Please accept our thanks for valuable comments that increased the value of the presented results. Due to the need to attach drawings, replies to comments were attached this time in the form of a separate file.

Yours faithfully

Anna Skwierawska 
